# Ladder Variational Autoencoders

**Casper Kaae Sønderby**[*]
casperkaae@gmail.com

**Tapani Raiko**[†]
tapani.raiko@aalto.fi

**Lars Maaløe**[‡]
larsma@dtu.dk

**Søren Kaae Sønderby**[*]
skaaesonderby@gmail.com

**Ole Winther**[*,‡]
olwi@dtu.dk

## Abstract

Variational autoencoders are powerful models for unsupervised learning. However deep models with several layers of dependent stochastic variables are difficult to train which limits the improvements obtained using these highly expressive models. We propose a new inference model, the Ladder Variational Autoencoder, that recursively corrects the generative distribution by a data dependent approximate likelihood in a process resembling the recently proposed Ladder Network. We show that this model provides state of the art predictive log-likelihood and tighter log-likelihood lower bound compared to the purely bottom-up inference in layered Variational Autoencoders and other generative models. We provide a detailed analysis of the learned hierarchical latent representation and show that our new inference model is qualitatively different and utilizes a deeper more distributed hierarchy of latent variables. Finally, we observe that batch-normalization and deterministic warm-up (gradually turning on the KL-term) are crucial for training variational models with many stochastic layers.

## 1 Introduction

The recently introduced variational autoencoder (VAE) [10, 19] provides a framework for deep generative models. In this work we study how the variational inference in such models can be improved while not changing the generative model. We introduce a new inference model using the same top-down dependency structure in both the inference and generative models achieving state-of-the-art generative performance.

VAEs, consisting of hierarchies of conditional stochastic variables, are highly expressive models retaining the computational efficiency of fully factorized models, Figure 1 a). Although highly flexible these models are difficult to optimize for deep hierarchies due to multiple layers of conditional stochastic layers. The VAEs considered here are trained by optimizing a variational approximate posterior lower bounding the intractable true posterior. Recently used inference are calculated purely bottom-up with no interaction between the inference and generative models [10, 18, 19]. We propose a new structured inference model using the same top-down dependency structure in both the inference and generative models. Here the approximate posterior distribution can be viewed as merging information from a bottom up computed approximate likelihood term with top-down prior information from the generative distribution, see Figure 1 b). The sharing of information (and parameters) with the generative model gives the inference model knowledge of the current state of the generative model in each layer. The top down-pass then recursively corrects the generative distribution with a data dependent approximating the log-likelihood using a simple precision-weighted addition.

---

[*]Bioinformatics Centre, Department of Biology, University of Copenhagen, Denmark

[†]Department of Computer Science, Aalto University, Finland

[‡]Department of Applied Mathematics and Computer Science, Technical University of Denmark

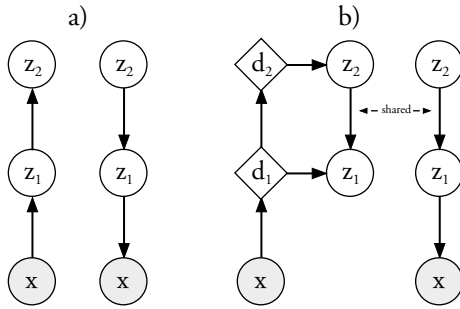

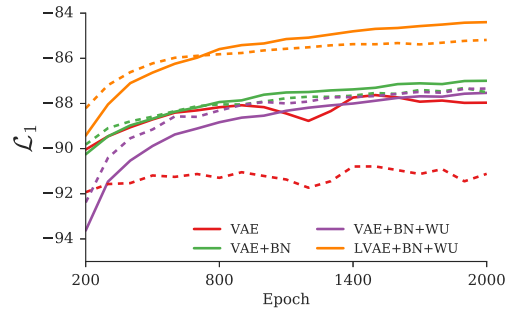

Figure 1: Inference (or encoder/recognition) and generative (or decoder) models for a) VAE and b) LVAE. Circles are stochastic variables and diamonds are deterministic variables.

Figure 2: MNIST train (*full lines*) and test (*dashed lines*) set log-likelihood using one importance sample during training. The LVAE improves performance significantly over the regular VAE.

This parameterization allows interactions between the *bottom-up* and *top-down* signals resembling the recently proposed Ladder Network [22, 17], and we therefore denote it Ladder-VAE (LVAE). For the remainder of this paper we will refer to VAEs as both the inference and generative model seen in Figure 1 a) and similarly LVAE as both the inference and generative model in Figure 1 b). We stress that the VAE and LVAE have identical generative models and only differ in the inference models.

Previous work on VAEs have been restricted to shallow models with one or two layers of stochastic latent variables. The performance of such models are constrained by the restrictive mean field approximation to the intractable posterior distribution. Here we found that purely bottom-up inference optimized with gradient ascent are only to a limited degree able to utilize more than two layers of stochastic latent variables. We initially show that a warm-up period [2, 16, Section 6.2] to support stochastic units staying active in early training and batch-normalization (BN) [7] can significantly improve performance of VAEs. Using these VAE models as competitive baselines we show that LVAE improves the generative performance achieving as good or better performance than other (often complicated) methods for creating flexible variational distributions such as: The Variational Gaussian Processes [21], Normalizing Flows [18], Importance Weighted Autoencoders [3] or Auxiliary Deep Generative Models[13]. Compared to the bottom-up inference in VAEs we find that LVAE: 1) have better generative performance 2) provides a tighter bound on the true log-likelihood and 3) can utilize deeper and more distributed hierarchies of stochastic variables. Lastly we study the learned latent representations and find that these differ qualitatively between the LVAE and VAE with the LVAE capturing more high level structure in the datasets. In summary our contributions are:

- A new inference model combining a Gaussian term, akin to an approximate Gaussian likelihood, with the generative model resulting in better generative performance than the normally used bottom-up VAE inference.

- We provide a detailed study of the learned latent distributions and show that LVAE learns both a deeper and more distributed representation when compared to VAE.

- We show that a deterministic warm-up period and batch-normalization are important for training deep stochastic models.

## 2  Methods

VAEs and LVAEs simultaneously train a generative model $p_\theta(\mathbf{x}, \mathbf{z}) = p_\theta(\mathbf{x}|\mathbf{z})p_\theta(\mathbf{z})$ for data $\mathbf{x}$ using latent variables $\mathbf{z}$, and an inference model $q_\phi(\mathbf{z}|\mathbf{x})$ by optimizing a variational lower bound to the likelihood $p_\theta(\mathbf{x}) = \int p_\theta(\mathbf{x}, \mathbf{z})d\mathbf{z}$. In the generative model $p_\theta$, the latent variables $\mathbf{z}$ are split into $L$ layers $\mathbf{z}_i$, $i = 1 \ldots L$, and each stochastic layer is a fully factorized Gaussian distribution conditioned

on the layer above:

$$p_\theta(\mathbf{z}) = p_\theta(\mathbf{z}_L) \prod_{i=1}^{L-1} p_\theta(\mathbf{z}_i|\mathbf{z}_{i+1}) \qquad (1)$$

$$p_\theta(\mathbf{z}_i|\mathbf{z}_{i+1}) = \mathcal{N}\left(\mathbf{z}_i|\mu_{p,i}(\mathbf{z}_{i+1}), \sigma_{p,i}^2(\mathbf{z}_{i+1})\right), \quad p_\theta(\mathbf{z}_L) = \mathcal{N}\left(\mathbf{z}_L|\mathbf{0}, \mathbf{I}\right) \qquad (2)$$

$$p_\theta(\mathbf{x}|\mathbf{z}_1) = \mathcal{N}\left(\mathbf{x}|\mu_{p,0}(\mathbf{z}_1), \sigma_{p,0}^2(\mathbf{z}_1)\right) \text{ or } P_\theta(\mathbf{x}|\mathbf{z}_1) = \mathcal{B}\left(\mathbf{x}|\mu_{p,0}(\mathbf{z}_1)\right) \qquad (3)$$

where the observation model is matching either continuous-valued (Gaussian $\mathcal{N}$) or binary-valued (Bernoulli $\mathcal{B}$) data, respectively. We use subscript $p$ (and $q$) to highlight if $\mu$ or $\sigma^2$ belongs to the generative or inference distributions respectively. Note that while individual conditional distributions are fully factorized, the hierarchical specification allows the lower layers of the latent variables to be highly correlated. The variational principle provides a tractable lower bound on the log likelihood which can be used as a training criterion $\mathcal{L}$.

$$\log p(\mathbf{x}) \geq E_{q_\phi(z|x)}\left[\log \frac{p_\theta(\mathbf{x}, \mathbf{z})}{q_\phi(\mathbf{z}|\mathbf{x})}\right] = \mathcal{L}(\theta, \phi; \mathbf{x}) \qquad (4)$$

$$= -KL(q_\phi(\mathbf{z}|\mathbf{x})||p_\theta(\mathbf{z})) + E_{q_\phi(\mathbf{z}|\mathbf{x})}\left[\log p_\theta(\mathbf{x}|\mathbf{z})\right], \qquad (5)$$

where $KL$ is the Kullback-Leibler divergence. A strictly tighter bound on the likelihood may be obtained at the expense of a $K$-fold increase of samples by using the importance weighted bound [3]:

$$\log p(\mathbf{x}) \geq E_{q_\phi(\mathbf{z}^{(1)}|\mathbf{x})} \ldots E_{q_\phi(\mathbf{z}^{(K)}|\mathbf{x})}\left[\log \sum_{k=1}^{K} \frac{p_\theta(\mathbf{x}, \mathbf{z}^{(k)})}{q_\phi(\mathbf{z}^{(k)}|\mathbf{x})}\right] = \mathcal{L}_K(\theta, \phi; \mathbf{x}) \geq \mathcal{L}(\theta, \phi; \mathbf{x}). \quad (6)$$

The generative and inference parameters, $\theta$ and $\phi$, are jointly trained by optimizing Eq. (5) using stochastic gradient descent where we use the reparametrization trick for stochastic backpropagation through the Gaussian latent variables [10, 19]. The $KL[q_\phi|p_\theta]$ is calculated analytically at each layer when possible and otherwise approximated using Monte Carlo sampling.

## 2.1 Variational autoencoder inference model

VAE inference models are parameterized as a bottom-up process similar to [3, 9]. Conditioned on the stochastic layer below each stochastic layer is specified as a fully factorized Gaussian distribution:

$$q_\phi(\mathbf{z}|\mathbf{x}) = q_\phi(\mathbf{z}_1|\mathbf{x}) \prod_{i=2}^{L} q_\phi(\mathbf{z}_i|\mathbf{z}_{i-1}) \qquad (7)$$

$$q_\phi(\mathbf{z}_1|\mathbf{x}) = \mathcal{N}\left(\mathbf{z}_1|\mu_{q,1}(\mathbf{x}), \sigma_{q,1}^2(\mathbf{x})\right) \qquad (8)$$

$$q_\phi(\mathbf{z}_i|\mathbf{z}_{i-1}) = \mathcal{N}\left(\mathbf{z}_i|\mu_{q,i}(\mathbf{z}_{i-1}), \sigma_{q,i}^2(\mathbf{z}_{i-1})\right), \, i = 2 \ldots L. \qquad (9)$$

In this parameterization the inference and generative distributions are computed separately with no explicit sharing of information. In the beginning of the training procedure this might cause problems since the inference models have to approximately match the highly variable generative distribution in order to optimize the likelihood. The functions $\mu(\cdot)$ and $\sigma^2(\cdot)$ in the generative and VAE inference models are implemented as:

$$\mathbf{d}(\mathbf{y}) = \texttt{MLP}(\mathbf{y}) \qquad (10)$$

$$\mu(\mathbf{y}) = \texttt{Linear}(\mathbf{d}(\mathbf{y})) \qquad (11)$$

$$\sigma^2(\mathbf{y}) = \texttt{Softplus}(\texttt{Linear}(\mathbf{d}(\mathbf{y}))), \qquad (12)$$

where $\texttt{MLP}$ is a two layered multilayer perceptron network, $\texttt{Linear}$ is a single linear layer, and $\texttt{Softplus}$ applies $\log(1 + \exp(\cdot))$ nonlinearity to each component of its argument vector ensuring positive variances. In our notation, each $\texttt{MLP}(\cdot)$ or $\texttt{Linear}(\cdot)$ gives a new mapping with its own parameters, so the deterministic variable $\mathbf{d}$ is used to mark that the $\texttt{MLP}$-part is shared between $\mu$ and $\sigma^2$ whereas the last $\texttt{Linear}$ layer is not shared.

## 2.2 Ladder variational autoencoder inference model

We propose a new inference model that recursively corrects the generative distribution with a data dependent approximate likelihood term. First a deterministic upward pass computes the Gaussian

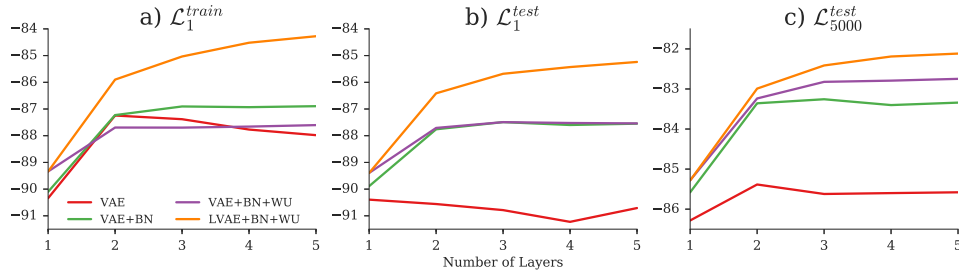

Figure 3: MNIST log-likelihood values for VAEs and the LVAE model with different number of latent layers, Batch-normalization (*BN*) and Warm-up (*WU*). a) Train log-likelihood, b) test log-likelihood and c) test log-likelihood with 5000 importance samples.

likelihood like contributions:

$$\mathbf{d}_n = \texttt{MLP}(\mathbf{d}_{n-1}) \tag{13}$$

$$\hat{\mu}_{q,i} = \texttt{Linear}(\mathbf{d}_i), i = 1 \ldots L \tag{14}$$

$$\hat{\sigma}^2_{q,i} = \texttt{Softplus}(\texttt{Linear}(\mathbf{d}_i)), i = 1 \ldots L \tag{15}$$

where $\mathbf{d}_0 = \mathbf{x}$. This is followed by a stochastic downward pass recursively computing both the approximate posterior and generative distributions:

$$q_\phi(\mathbf{z}|\mathbf{x}) = q_\phi(\mathbf{z}_L|\mathbf{x}) \prod_{i=1}^{L-1} q_\phi(\mathbf{z}_i|\mathbf{z}_{i+1}, \mathbf{x}) \tag{16}$$

$$\sigma_{q,i} = \frac{1}{\hat{\sigma}^{-2}_{q,i} + \sigma^{-2}_{p,i}} \tag{17}$$

$$\mu_{q,i} = \frac{\hat{\mu}_{q,i}\hat{\sigma}^{-2}_{q,i} + \mu_{p,i}\sigma^{-2}_{p,i}}{\hat{\sigma}^{-2}_{q,i} + \sigma^{-2}_{p,i}} \tag{18}$$

$$q_\phi(\mathbf{z}_i|\cdot) = \mathcal{N}\left(\mathbf{z}_i|\mu_{q,i}, \sigma^2_{q,i}\right), \tag{19}$$

where $\mu_{q,L} = \hat{\mu}_{q,L}$ and $\sigma^2_{q,L} = \hat{\sigma}^2_{q,L}$. The inference model is a precision-weighted combination of $\hat{\mu}_q$ and $\hat{\sigma}^2_q$ carrying bottom-up information and $\mu_p$ and $\sigma^2_p$ from the generative distribution carrying *top-down* prior information. This parameterization has a probabilistic motivation by viewing $\hat{\mu}_q$ and $\hat{\sigma}^2_q$ as an approximate Gaussian likelihood that is combined with a Gaussian prior $\mu_p$ and $\sigma^2_p$ from the generative distribution. Together these form the approximate posterior distribution $q_\theta(\mathbf{z}|\mathbf{x})$ using the same top-down dependency structure both in the inference and generative model. A line of motivation, already noted in [4], see [1] for a recent approach, is that a purely bottom-up inference process as in i.e. VAEs does not correspond well with real perception, where iterative interaction between bottom-up and top-down signals produces the final activity of a unit[4]. Notably it is difficult for the purely bottom-up inference networks to model the *explaining away* phenomenon, see [23, Chapter 5] for a recent discussion on this phenomenon. The LVAE model provides a framework with the wanted interaction, while not increasing the number of parameters.

### 2.3 Warm-up from deterministic to variational autoencoder

The variational training criterion in Eq. (5) contains the reconstruction term $p_\theta(\mathbf{x}|\mathbf{z})$ and the variational regularization term. The variational regularization term causes some of the latent units to become uninformative during training [14] because the approximate posterior for unit $k$, $q(z_{i,k}|\ldots)$ is regularized towards its own prior $p(z_{i,k}|\ldots)$, a phenomenon also recognized in the VAE setting [3, 2]. This can be seen as a virtue of automatic relevance determination, but also as a problem when many units collapse early in training before they learned a useful representation. We observed that such units remain uninformative for the rest of the training, presumably trapped in a local minima or saddle point at $KL(q_{i,k}|p_{i,k}) \approx 0$, with the optimization algorithm unable to re-activate them.

We alleviate the problem by initializing training using the reconstruction error only (corresponding to training a standard deterministic auto-encoder), and then gradually introducing the variational regularization term:

$$\mathcal{L}(\theta, \phi; \mathbf{x})_{WU} = -\beta KL(q_\phi(z|x)||p_\theta(\mathbf{z})) + E_{q_\phi(z|x)}\left[\log p_\theta(\mathbf{x}|\mathbf{z})\right], \qquad (20)$$

where $\beta$ is increased linearly from 0 to 1 during the first $N_t$ epochs of training. We denote this scheme *warm-up* (abbreviated *WU* in tables and graphs) because the objective goes from having a delta-function solution (corresponding to zero temperature) and then move towards the fully stochastic variational objective. This idea have previously been considered in [16, Section 6.2] and more recently in [2].

## 3 Experiments

To test our models we use the standard benchmark datasets MNIST, OMNIGLOT [11] and NORB [12]. The largest models trained used a hierarchy of five layers of stochastic latent variables of sizes 64, 32, 16, 8 and 4, going from bottom to top. We implemented all mappings using MLP's with two layers of deterministic hidden units. In all models the MLP's between $x$ and $z_1$ or $d_1$ were of size 512. Subsequent layers were connected by MLP's of sizes 256, 128, 64 and 32 for all connections in both the VAE and LVAE. Shallower models were created by removing latent variables from the top of the hierarchy. We sometimes refer to the five layer models as 64-32-16-8-4, the four layer models as 64-32-16-8 and so fourth. The models were trained end-to-end using the Adam [8] optimizer with a mini-batch size of 256. We report the train and test log-likelihood lower bounds, Eq. (5) as well as the approximated true log-likelihood calculated using 5000 importance weighted samples, Eq. (6). The models were implemented using the Theano [20], Lasagne [5] and Parmesan[5] frameworks. The source code is available at github[6]

For MNIST, we used a sigmoid output layer to predict the mean of a Bernoulli observation model and leaky rectifiers ($\max(x, 0.1x)$) as nonlinearities in the MLP's. The models were trained for 2000 epochs with a learning rate of 0.001 on the complete training set. Models using warm-up used $N_t = 200$. Similarly to [3], we resample the binarized training values from the real-valued images using a Bernoulli distribution after each epoch which prevents the models from over-fitting. Some of the models were fine-tuned by continuing training for 2000 epochs while multiplying the learning rate with 0.75 after every 200 epochs and increase the number of Monte Carlo and importance weighted samples to 10 to reduce the variance in the approximation of the expectations in Eq. (4) and improve the inference model, respectively.

Models trained on the OMNIGLOT dataset[7], consisting of 28x28 binary images images were trained similar to above except that the number of training epochs was 1500.

Models trained on the NORB dataset[8], consisting of 32x32 grays-scale images with color-coding rescaled to $[0, 1]$, used a Gaussian observation model with mean and variance predicted using a linear and a softplus output layer respectively. The settings were similar to the models above except that *hyperbolic tangent* was used as nonlinearities in the MLP's and the number of training epochs was 2000.

### 3.1 Generative log-likelihood performance

In Figure 3 we show the train and test set log-likelihood on the MNIST dataset for a series of different models with varying number of stochastic layers.

Consider the $\mathcal{L}_1^{test}$, Figure 3 b), the VAE without batch-normalization and warm-up does not improve for additional stochastic layers beyond one whereas VAEs with batch-normalization and warm-up improve performance up to three layers. The LVAE models performs better improving performance for each additional layer reaching $\mathcal{L}_1^{test} = -85.23$ with five layers which is significantly higher than the best VAE score at $-87.49$ using three layers. As expected the improvement in performance is

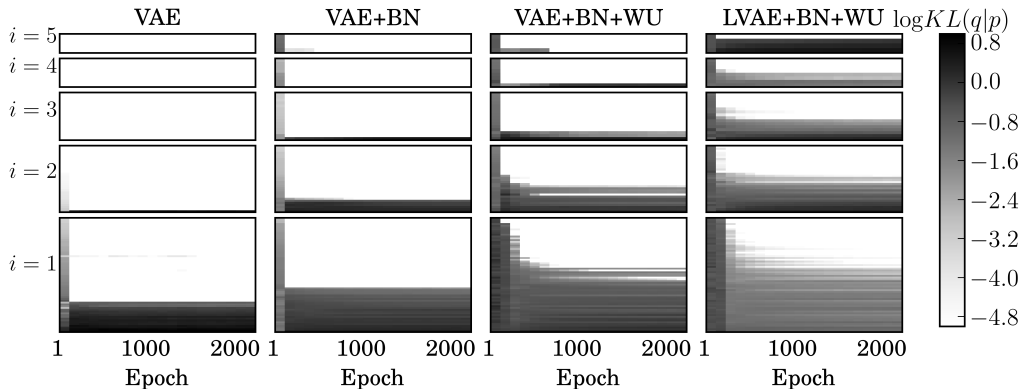

Figure 4: $\log KL(q|p)$ for each latent unit is shown at different training epochs. Low $KL$ (white) corresponds to an uninformative unit. The units are sorted for visualization. It is clear that vanilla VAE cannot train the higher latent layers, while introducing batch-normalization helps. Warm-up creates more active units early in training, some of which are then gradually pruned away during training, resulting in a more distributed final representation. Lastly, we see that the LVAE activates the highest number of units in each layer.

|  | $\leq \log p((x))$ |
| --- | --- |
| VAE 1-layer + NF [18] | -85.10 |
| IWAE, 2-layer + IW=1 [3] | -85.33 |
| IWAE, 2-layer + IW=50 [3] | -82.90 |
| VAE, 2-layer + VGP [21] | -81.90 |
| LVAE, 5-layer | -82.12 |
| LVAE, 5-layer + finetuning | -81.84 |
| LVAE, 5-layer + finetuning + IW=10 | -81.74 |

Table 1: Test set MNIST performance for importance weighted autoencoder (IWAE), VAE with normalizing flows (NF) and VAE with variational Gaussian process (VGP). Number of importance weighted (IW) samples used for training is one unless otherwise stated.

decreasing for each additional layer, but we emphasize that the improvements are consistent even for the addition of the top-most layers. We found batch-normalization improved performance for all models, however especially for LVAE we found batch-normalization to be important. In Figure 3 c) the approximated true log-likelihood estimated using 5000 importance weighted samples is seen. Again the LVAE models performs better than the VAE reaching $\mathcal{L}_{5000}^{test} = -82.12$ compared to the best VAE at $-82.74$. These results show that the LVAE achieves both a higher approximate log-likelihood score, but also a significantly tighter lower bound on the log-likelihood $\mathcal{L}_1^{test}$. The models in Figure 3 were trained using fixed learning rate and one Monte Carlo and importance weighted sample. To improve performance we fine-tuned the best performing five layer LVAE models by training these for a further 2000 epochs with annealed learning rate and increasing the number of IW samples and see a slight improvements in the test set log-likelihood values, Table 1. We saw no signs of over-fitting for any of our models even though the hierarchical latent representations are highly expressive as seen in Figure 2.

Comparing the results obtained here with current state-of-the art results on permutation invariant MNIST, Table 1, we see that the LVAE performs better than the normalizing flow VAE and importance weighted VAE and comparable to the Variational Gaussian Process VAE. However we note that these results are not directly comparable to these due to differences in the training procedure.

To test the models on more challenging data we used the OMNIGLOT dataset, consisting of characters from 50 different alphabets with 20 samples of each character. The log-likelihood values, Table 2, shows similar trends as for MNIST with the LVAE achieving the best performance using five layers

|              | VAE     | VAE +BN | VAE +BN +WU | LVAE +BN +WU |
|--------------|---------|---------|-------------|--------------|
| **OMNIGLOT** |         |         |             |              |
| 64           | −111.21 | −105.62 | −104.51     | −            |
| 64-32        | −110.58 | −105.51 | −102.61     | −102.63      |
| 64-32-16     | −111.26 | −106.09 | −102.52     | −102.18      |
| 64-32-16-8   | −111.58 | −105.66 | −102.66     | −102.21      |
| 64-32-16-8-4 | −110.46 | −105.45 | −102.48     | **-102.11**  |
|              |         |         |             |              |
| **NORB**     |         |         |             |              |
| 64           | 2741    | 3198    | 3338        | −            |
| 64-32        | 2792    | 3224    | 3483        | 3272         |
| 64-32-16     | 2786    | 3235    | 3492        | **3519**     |
| 64-32-16-8   | 2689    | 3201    | 3482        | 3449         |
| 64-32-16-8-4 | 2654    | 3198    | 3422        | 3455         |

Table 2: Test set log-likelihood scores for models trained on the OMNIGLOT and NORB datasets. The left most column show dataset and the number of latent variables i each model.

of latent variables, see the appendix for further results. The best log-likelihood results obtained here, $−102.11$, is higher than the best results from [3] at $−103.38$, which were obtained using more latent variables (100-50 vs 64-32-16-8-4) and further using 50 importance weighted samples for training.

We tested the models using a continuous Gaussian observation model on the NORB dataset consisting of gray-scale images of 5 different toy objects under different illuminations and observation angles. The LVAE achieves a slightly higher score than the VAE, however none of the models see an increase in performance for more using more than three stochastic layers. We found the Gaussian observation models to be harder to optimize compared to the Bernoulli models, a finding also recognized in [24], which might explain the lower utilization of the topmost latent layers in these models.

## 3.2  Latent representations

The probabilistic generative models studied here automatically tune the model complexity to the data by reducing the effective dimension of the latent representation due to the regularization effect of the priors in Eq. (4). However, as previously identified [16, 3], the latent representation is often overly sparse with few stochastic latent variables propagating useful information.

To study the importance of individual units, we split the variational training criterion $\mathcal{L}$ into a sum of terms corresponding to each unit $k$ in each layer $i$. For stochastic latent units, this is the KL-divergence between $q(z_{i,k}|\cdot)$ and $p(z_{i,k}|\mathbf{z}_{i+1})$. Figure 4 shows the evolution of these terms during training. This term is zero if the inference model is collapsed onto the prior carrying no information about the data, making the unit uninformative. For the models without warm-up we find that the KL-divergence for each unit is stable during all training epochs with only very few new units activated during training. For the models trained with warm-up we initially see many active units which are then gradually pruned away as the variational regularization term is introduced. At the end of training warm-up results in more active units indicating a more distributed representation and further that the LVAE model produces both the deepest and most distributed latent representation.

We also study the importance of layers by splitting the training criterion layer-wise as seen in Figure 5. This measures how much of the representation work (or innovation) is done in each layer. The VAEs use the lower layers the most whereas the highest layers are not (or only to a limited degree) used. Contrary to this, the LVAE puts much more importance to the higher layers which shows that it learns both a deeper and qualitatively different hierarchical latent representation which might explain the better performance of the model. To qualitatively study the learned representations, PCA plots of $\mathbf{z}_i \sim q(\mathbf{z}_i|\cdot)$ are seen in Figure 6. For vanilla VAE, the latent representations above the second layer are completely collapsed on a standard normal prior. Including Batch-normalization and warm-up activates one additional layer each in the VAE. The LVAE utilizes all five latent layers and the latent representation shows progressively more clustering according to class, which is clearly seen in the

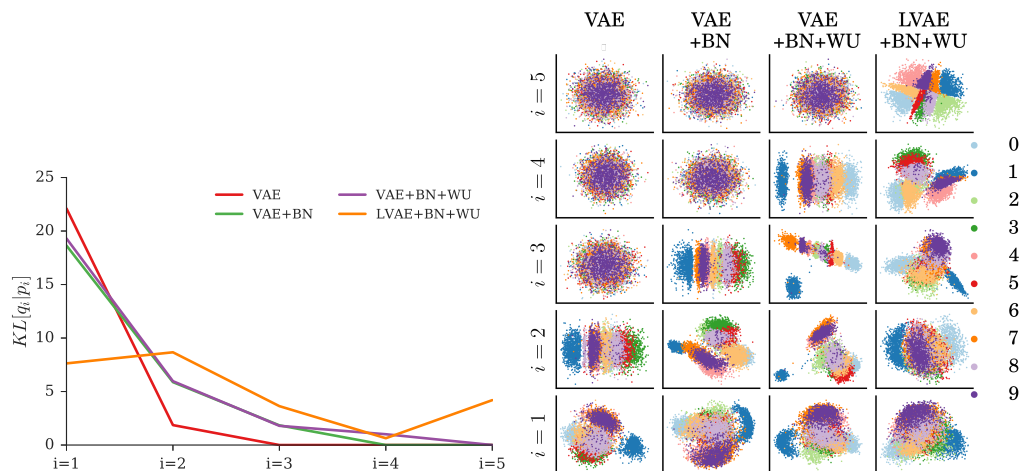

Figure 5: Layer-wise $KL[q|p]$ divergence going from the lowest to the highest layers. In the VAE models the KL divergence is highest in the lowest layers whereas it is more distributed in the LVAE model

Figure 6: PCA-plots of samples from $q(z_i|z_{i-1})$ for 5-layer VAE and LVAE models trained on MNIST. Color-coded according to true class label

topmost layer of this model. These findings indicate that the LVAE produce a structured high-level latent representations that are likely useful for semi-supervised learning.

## 4 Conclusion and Discussion

We presented a new inference model for VAEs combining a bottom-up data-dependent approximate likelihood term with prior information from the generative distribution. We showed that this parameterization 1) increases the approximated log-likelihood compared to VAEs, 2) provides a tighter bound on the log-likelihood and 3) learns a deeper and qualitatively different latent representation of the data. Secondly we showed that deterministic warm-up and batch-normalization are important for optimizing deep VAEs and LVAEs. Especially the large benefits in generative performance and depth of learned hierarchical representations using batch-normalization were surprising given the additional noise introduced. This is something that is not fully understood and deserves further investigation and although batch-normalization is not novel we believe that this finding in the context of VAEs are important.

The inference in LVAE is computed recursively by correcting the generative distribution with a data-dependent approximate likelihood contribution. Compared to purely bottom-up inference, this parameterization makes the optimization easier since the inference is simply correcting the generative distribution instead of fitting the two models separately. We believe this explicit parameter sharing between the inference and generative distribution can generally be beneficial in other types of recursive variational distributions such as DRAW [6] where the ideas presented here are directly applicable. Further the LVAE is orthogonal to other methods for improving the inference distribution such as Normalizing flows [18], Variational Gaussian Process [21] or Auxiliary Deep generative models [13] and combining with these might provide further improvements.

Other directions for future work include extending these models to semi-supervised learning which will likely benefit form the learned deep structured hierarchies of latent variables and studying more elaborate inference schemes such as a $k$-step iterative inference in the LVAE [15].

## Footnotes

[4]The idea was dismissed at the time, since it could introduce substantial theoretical complications.

[5]https://github.com/casperkaae/parmesan

[6]https://github.com/casperkaae/LVAE

[7]The OMNIGLOT data was partitioned and preprocessed as in [3], https://github.com/yburda/iwae/tree/master/datasets/OMNIGLOT

[8]The NORB dataset was downloaded in resized format from github.com/gwtaylor/convnet_matlab

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
