[Supplementary Material]

# Appendix: Ladder Variational Autoencoders

**Casper Kaae Sønderby**[*]
casperkaae@gmail.com

**Tapani Raiko**[†]
tapani.raiko@aalto.fi

**Lars Maaløe**[‡]
larsma@dtu.dk

**Søren Kaae Sønderby**[*]
skaaesonderby@gmail.com

**Ole Winther**[*,‡]
olwi@dtu.dk

Figure 1: MNIST log-likelihood values for VAEs and the LVAE model with different number of latent layers, Batch normalization (*BN*) and Warm-up (*WU*). a) Train log-likelihood, b) test log-likelihood and c) test log-likelihood with 5000 importance samples. Note that the LVAE without batch normalization performed very poorly why some of the results fall outside the range of the plots

Figure 2: OMNIGLOT log-likelihood values for VAEs and the LVAE model with different number of latent layers, Batch normalization (*BN*) and Warm-up (*WU*). a) Train log-likelihood, b) test log-likelihood and c) test log-likelihood with 5000 importance samples

[*]Bioinformatics Centre, Department of Biology, University of Copenhagen, Denmark

[†]Department of Computer Science, Aalto University, Finland

[‡]Department of Applied Mathematics and Computer Science, Technical University of Denmark

Figure 3: MNIST samples. a) True data, b) Conditional Reconstructions and c) Samples from the prior distribution

Figure 4: MNIST samples. a) True data, b) Conditional Reconstructions and c) Samples from the prior distribution