[Reviews · NeurIPS 2016]

Reviewer 1

Summary

This paper explores three techniques for training variational autoencoders (VAEs) with multiple stochastic layers more effectively. The most interesting of these involves a novel parameterization of the inference network that performs a deterministic bottom-up pass followed by a stochastic top-down pass defined by the conditional mappings from the model prior. The other two techniques are using batch normalization (BN) in the VAE and warming up the latent units (WU) by annealing the weight on the KL term in the variational objective from 0 to 1 to encourage the model to use more latent units. The experimental results on MNIST and Omniglot (and to a lesser degree NORB) indicate that all three techniques result in VAEs that use more latent units, especially in deeper layers. The best results are obtained by using all three techniques together.

Qualitative Assessment

This is an interesting and fairly well written paper. The new inference network parameterization inspired by Ladder Networks makes intuitive sense though I would have liked a stronger motivation for precise way the bottom-up and top-down information is combined (Eq. 17 and 18). Have the authors explored alternative (e.g. simpler) ways of combining the two sources of information? If so, reporting the less successful parameterizations considered would make the paper stronger. An explanation why the bottom-up contribution (mu-hat and sigma-hat) can be seen as an approximate likelihood (l. 102) would also be welcome. The experimental section is informative and thorough. The only thing I thought was missing were the results for LVAEs trained using only WU or BN (but not both). Given the shortage of space, a sentence or two describing what happens in this case would be sufficient. Calling a unit with the posterior matching the prior "inactive" (l. 201) seems misleading as it is not constant. Something like "uninformative" or "disconnected" might be better. Finally, the paper would benefit from more proofreading as there is a number of typos remaining.

Confidence in this Review

3-Expert (read the paper in detail, know the area, quite certain of my opinion)


Reviewer 2

Summary

In this paper the authors propose a method to combine the (parameters of) the approximate inference model Q and the generative model P into a better approximate inference method for variational autoencoder-style models. The basic idea is to run the deterministic part of the bottom-up inference model and top-down generative model and combine their predictions into a joint prediction before drawing samples.

Qualitative Assessment

Besides of introducing and evaluating the core idea behind this paper, which is to combine the bottom-up and top down models into a inference method that contains both, this paper paper also provides an analysis of two techniques to improve the optimization for VAE-style models: batch-normalisation and warm-up. The experimental section contains an analysis that nicely disentangles the effects of these techniques. In my own (unpublished) experiments I observe that VAE models that were trained with batch-normalization often lead to “worse looking samples” (judged by a human observer). I wonder whether the authors observe a similar effect? Nevertheless, in general I find the experiments and their analysis convincing. On the theoretical side, the authors might want to cite the “Bidirectional Helmholtz Machines” paper. Similar to this work, the BiHM paper also proposes to combine the P and Q models into a joint model and reported that such models prefer significantly deeper architectures -- a result that is also prominently mentioned in this paper.

Confidence in this Review

3-Expert (read the paper in detail, know the area, quite certain of my opinion)


Reviewer 3

Summary

The authors propose a modification of the variational auto-encoder that consists in correcting the model's generative distribution with a data dependent likelihood factors, in a process that resembles the one used by the ladder network: a non-probabilistic auto-encoder that tries to match the values of the hidden layers in the encoder and decoder networks. The proposed modification is based on using a deterministic inference network in the variational auto-encoder that samples the top latent variables from a Gaussian distribution, as in the variational auto-encoder, after that the latent variables at intermediate layers are sampled from a product of Gaussian factors, where one of the factors come from the original generative model, and the others come from the inference network. The experiments show that the proposed modification results in better estimates of the test-log-likelihood. The proposed model learns richer latent distributions. The experiments performed also show that using batch normalization and a deterministic warm-up approach during training are important for obtaining improved results.

Qualitative Assessment

Quality: the paper seems to be technically sound. However, I do not like that the authors report results on table 1 that are not directly comparable because of differences in the training procedure. Clarity: the paper is clearly written. However, the authors could have done a much better job at explaining how the inference network generates values for the latent variables. I took me too long to fully understand the whole process. It is not clear how the authors apply the batch normalization procedure during training. They should describe this in the text. I assume they only use batch-normalization in the inference network since using it in the generative network would change the generative model. What are the discontinuous lines in Figure 2? They should be described in the caption of the figure. Originality: the contribution is original. The authors proposed the first probabilistic equivalent of the ladder network, a model that has been very successful. They also provide significant insights on how to improve the VAE by using sensible modifications for the training process by using warm-ups and batch normalization. Significance: the experimental results clearly show the advantages of the proposed approach. The modifications are therefore significant. However, it still not clear why this approach should be preferred with respect to other possible alternatives.

Confidence in this Review

2-Confident (read it all; understood it all reasonably well)


Reviewer 4

Summary

The authors introduce a 'ladder variational autoencoder', which enriches the VAE family by a model inspired by the ladder networks literature of Harri Valpola (i.e. 'Neural PCA...') and introduces skip connections between the states of the inference and generation pathways. The authors show that using this model with a deterministic inference they can learn models with higher likelihoods in benchmark datasets as well as deeper VAEs than currently commonly feasible. They discuss a trick they call 'warm up' which is a simple annealing framework to help the optimization of the energy landscape in their model.

Qualitative Assessment

The paper is solid, but has a few issues. The authors argue that the introduction of their skip connections models top down and bottom-up inference, yet do not show any results supporting that claim. Furthermore, the details of the variational approximation and the implication on the model by the introduction of these skip connections are heavily underexplored, to the point of being un-understandable. I would worry about short-circuiting their VAEs in a non-trivial way without a deeper mathematical understanding of the skip connections. On the positive side, the paper shows that there is a mode in which these connections work and appear to do something useful, as evidenced by the learning stability allowing deeper VAEs and the shape of the latent spaces which seems to be semantically structured. The likelihood results are hard to trust without the precise corresponding model equations showing how these differences can be explained. In addition, recently published models like the Variational GP (Ranganath, Tran Blei 2016) outperform the LVAEs heavily in terms of likelihood.

Confidence in this Review

3-Expert (read the paper in detail, know the area, quite certain of my opinion)


Reviewer 5

Summary

The paper integrates ladder architecture, KL annealing, and batch normalization to improve variational autoencoders with multiple stochastic layers.

Qualitative Assessment

The paper is nicely written and carefully details its model and experiments. The intuition behind the combination of a bottom-up, top-down inference is interesting, though I believe the authors overstate their point when they say VAE can 'only' do pure bottom-up inference. From a computational graph prospective, what the authors propose in fine is a complex network which samples the latent variables top-down, with a fair amount of skip connections (the lateral connections), hidden units with the semantic of mean and standard deviation, shared parameters between the generative model and the inference network, and some parameterless, differentiable layers (the 'combination' layers of equation 18-19). It has a very nice interpretation as the authors suggest. But computationally, nothing prevents a complex inference network with top-down sampling (compatible with classical VAE framework) to carry 'bottom up' in its information in its hidden units and effectively perform similar computation. In other words, this particular model is in the model class of a sufficiently complex inference network (which admittedly could be very hard to learn). The author effectively constrain the inference network to have a particular inductive bias - a generally good idea! Other than that, I don't have major comments. The idea themselves are possibly not too surprising, but their combination is novel and the results are strong. The paper would make a valuable addition to NIPS. Minor: - In figure 2, it is a bit surprising that VAE+BN+WU is worse than VAE+BN (even though it seems to generalize better). Do the authors have any intuition for this result? - Given how close the curves are in general, adding confidence intervals around the estimates would have been desirable.

Confidence in this Review

3-Expert (read the paper in detail, know the area, quite certain of my opinion)


Reviewer 6

Summary

This paper presents a ladder variational auto-encoder framework, which resembles the structure of ladder network but builds the connection of latent variables in a different way. The experiments in unsupervised learning show state of art performance in some benchmark datasets.

Qualitative Assessment

The most technical part of this paper is to propose the precision-weighted combination of bottom-up and top-down information. This connection between latent variables in generative model and recognition model follows the computation of posterior for Gaussian distribution with known variance and conjugated Gaussian prior for parameter mean. This is the highlight of the paper, but the overall contribution is incremental. I have a few concerns or questions on this paper. 1. In Eq (16), q(z_i|z_{i+1}) should be q(z_i|z_{i+1}, x) or q(z_i|z_{i+1}, d_i(x))? 2. In Line 103, what does q_{\theta}(z|z,x) mean? 3. In Fig 3, is the result of VAE fine-tuned as LVAE, such as anneal learning rate? Because my previous experience told me VAE with 1 stochastic layer (embedded by 2-layer MLP) can roughly achieve -85. 4. I'd like to see the results of vanilla LVAE. Like the "deconstructing the ladder networks", I want to understand how the proposed construction of information flow or together with other tricks (BN, WU) contributes to the model performance. 5. I think the results of unsupervised learning is not enough, since the lower bound of VAE (-82.74) with WU+BN is close to LVAE (-82.12). I would like to see the performance of LVAE in semi-supervised training. The success of ladder networks is nominated for this task. 6. In addition, similar lower bound may not necessarily indicate similar reconstruction error. I used to observe very different reconstruction errors (in the sense of pixel-wise MSE) under similar lower bound.

Confidence in this Review

3-Expert (read the paper in detail, know the area, quite certain of my opinion)